# The RNA-Binding Landscape of HAX1 Protein Indicates Its Involvement in Translation and Ribosome Assembly

**DOI:** 10.3390/cells11192943

**Published:** 2022-09-20

**Authors:** Anna Balcerak, Ewelina Macech-Klicka, Maciej Wakula, Rafal Tomecki, Krzysztof Goryca, Malgorzata Rydzanicz, Mateusz Chmielarczyk, Malgorzata Szostakowska-Rodzos, Marta Wisniewska, Filip Lyczek, Aleksandra Helwak, David Tollervey, Grzegorz Kudla, Ewa A. Grzybowska

**Affiliations:** 1Molecular and Translational Oncology, Maria Sklodowska-Curie National Research Institute of Oncology, 02-781 Warsaw, Poland; 2Laboratory of RNA Processing and Decay, Institute of Biochemistry and Biophysics, Polish Academy of Sciences, 02-106 Warsaw, Poland; 3Faculty of Biology, Institute of Genetics and Biotechnology, University of Warsaw, 02-106 Warsaw, Poland; 4Genomics Core Facility, Centre of New Technologies University of Warsaw, 02-097 Warsaw, Poland; 5Department of Medical Genetics, Medical University of Warsaw, 02-106 Warsaw, Poland; 6Laboratory of Biological Chemistry of Metal Ions, Institute of Biochemistry and Biophysics, Polish Academy of Sciences, 02-106 Warsaw, Poland; 7Wellcome Centre for Cell Biology, University of Edinburgh, Edinburgh EH9 3BF, UK; 8MRC Human Genetics Unit, University of Edinburgh, Edinburgh EH4 2XU, UK

**Keywords:** RNA–protein binding, HAX1, RIP-seq, CRAC, translation, ribosome assembly

## Abstract

HAX1 is a human protein with no known homologues or structural domains. Mutations in the *HAX1* gene cause severe congenital neutropenia through mechanisms that are poorly understood. Previous studies reported the RNA-binding capacity of HAX1, but the role of this binding in physiology and pathology remains unexplained. Here, we report the transcriptome-wide characterization of HAX1 RNA targets using RIP-seq and CRAC, indicating that HAX1 binds transcripts involved in translation, ribosome biogenesis, and rRNA processing. Using CRISPR knockouts, we find that HAX1 RNA targets partially overlap with transcripts downregulated in *HAX1* KO, implying a role in mRNA stabilization. Gene ontology analysis demonstrated that genes differentially expressed in *HAX1* KO (including genes involved in ribosome biogenesis and translation) are also enriched in a subset of genes whose expression correlates with *HAX1* expression in four analyzed neoplasms. The functional connection to ribosome biogenesis was also demonstrated by gradient sedimentation ribosome profiles, which revealed differences in the small subunit:monosome ratio in *HAX1* WT/KO. We speculate that changes in *HAX1* expression may be important for the etiology of HAX1-linked diseases through dysregulation of translation.

## 1. Introduction

HAX1 (HCLS1-associated protein X-1) is known as an antiapoptotic protein with a role in the regulation of cell migration, cell adhesion, and calcium homeostasis [1]. HAX1 deficiency, due to mutations in the *HAX1* gene, results in the autosomal recessive severe congenital neutropenia (SCN) called Kostmann disease [2]. This myelopoietic disorder is caused by the arrest of granulocyte maturation and the resulting paucity of mature neutrophils, leading to life-threatening infections. This effect has been attributed to excessive apoptosis caused by HAX1 deficiency, but the exact molecular mechanism was not demonstrated. On the other hand, *HAX1* overexpression was documented in several neoplasms [3,4,5,6]. Here, we propose that these effects may be linked to ribosome dysfunction.

To date, HAX1’s RNA-binding propensity has been reported in two particular cases: for the vimentin transcript [7] and the DNA polymerase beta transcript [8]. Both instances pertained to strong hairpin structures at the 3′UTR of these transcripts, although the comparison of the hairpin motifs in vimentin and Pol β mRNAs did not reveal any significant similarities, apart from the presence of U-rich single-stranded regions [8]. In our recent report [9], in which we described the HAX1 protein interactome, we also suggested the possibility of RNA-binding by HAX1 deduced from neighboring proteins (first observed by Brannan et al. [10]).

The role of HAX1 RNA binding in cellular processes has not been clarified so far, except for some suggestions of its involvement in the regulation of specific mRNAs [11,12].

The HAX1 protein has no homologues or known domains, except for a PEST sequence, a specific signal which was linked to a shorter half-life of the protein and its proteasomal degradation [13]. The presence of BCL2-like domains was disproved [14]. Consequently, HAX1 does not possess any known RNA-binding domain, and a large proportion of the protein is predicted to be disordered [15,16], so RNA binding probably occurs in a nonconventional manner.

The current study provides for the first time a comprehensive analysis of the HAX1–RNA interactome, with two independent approaches for the isolation of its RNA targets. Subsequent analysis of the impact of HAX1 on the cell transcriptome and analysis of expression in several cancer databases produced coherent results that indicate the unanticipated role of HAX1 in ribosome biogenesis and translation. Comparison of the experimental data obtained for HAX1 RNA binding and transcriptome profiling indicates that HAX1 may regulate the stability of the bound transcripts. These results were corroborated by the observation that *HAX1* KO affects the ribosomal profile, especially with respect to the ratio of the small ribosomal subunit to the monosome. The involvement of HAX1 in ribosome biogenesis and translation emerging from this work may help to elucidate its many-sided effects on cellular processes and HAX1-associated diseases.

## 2. Results

### 2.1. Different High-Throughput Analyses of HAX1 Binding Targets Consistently Reveal Its Involvement in Translation and Ribosome Biogenesis

HAX1 is predicted to contain few secondary structure elements, while most of its sequence is predicted to be disordered and there is no conventional RNA-binding domain (Figure 1A, [15,16]). Nevertheless, the RNA-binding propensity of HAX1 was previously reported [7,8]. Furthermore, three different RNA-binding prediction algorithms suggest that HAX1 may bind to RNA: RNApred (amino acid composition-based method, the score for putative RBD: 0.58 SVM (support vector machine), cutoff: 0.5, RNApred. Available online: https://webs.iiitd.edu.in/raghava/rnapred/, accessed on 10 April 2019) [17], RBPPred (sequence-based method, the score for putative RBD: 0.928 SVM, cutoff: 0.5, RBPPred. Available online: http://rnabinding.com/RBPPred.html, accessed on 12 Februray 2020) [18], and CatRapid (secondary structure, hydrogen bonding, and van der Waals forces-based prediction, the score for putative RBD: 0.61, threshold 0.5, CatRapid. Available online: http://s.tartaglialab.com/page/catrapid_group, accessed on 21 March 2019) [19] (Figure 1B). Algorithms that compare three-dimensional structures cannot be used, since HAX1 has no homologs, is mostly disordered, and has not been crystalized.

To verify and globally characterize the RNA-binding propensity of HAX1, we used two independent, high-throughput screens (RIP-seq and CRAC) for the HAX1 RNA targets (Figure 1C,D). RIP-seq is simpler, with one purification step and without the crosslinking step, and thus it yields more background noise (>5000 hits). On the contrary, CRAC involved RNA crosslinking and two purification steps; thus, the results were more specific and enabled the analysis of the bound RNA regions. Both approaches unanimously revealed enrichment in the RNAs linked to translation, ribosome biogenesis, and RNA processing.

#### 2.1.1. RIP-Seq Results

RNA target identification was performed in the human leukemia cell line HL-60, because promyelocytes represent the most affected cell type in patients with a recessive mutation in both *HAX1* alleles (severe neutropenia). This condition is lethal when untreated, so the analysis of the role of HAX1 in these cells seems to be the most physiologically relevant. The experiment was carried out in five replicates for each, with a HAX1-IP and IgG control (correct clustering confirmed by principal component analysis, Appendix A). HAX1-associated transcriptome was identified by employing next generation sequencing after the RNA immunoprecipitation (RIP-seq) of the HAX1 complexes. Transcripts were classified as positive targets using an adjusted *p*-value cutoff of 0.05. Enrichment analysis (the Database for Annotation, Visualization and Integrated Discovery; DAVID) of the obtained dataset indicates involvement in RNA processing, translation, and ribosome biogenesis, including a strong category of targets involved in rRNA processing (Figure 2A, Appendix A). Using the DAVID Functional Classification Tool, which performs background compensation for *Homo sapiens* expression profile, ensures that the obtained pool of highly expressed transcripts encoding proteins involved in translation and ribosome biogenesis represents a truly significant output. Plotting enrichment against the false discovery rate (FDR) for GO terms obtained by gene ontology enrichment analysis (Panther) in the Biological Process category revealed that the terms related to ribosome (including ribosome biogenesis and rRNA processing) and translation are of high significance (high enrichment, low FDR, Figure 2B). Most of the RNA targets obtained in the RIP analysis represent mRNA (Figure 2C). Figure 2D represents the RIP-seq coverage (X-axis; FPKM, fragments per kilobase per million, to compensate for the length of the transcript) plotted against log2 enrichment in RIP-seq (RIP-seq_HAX1/RIP-seq_IgG). The red points above (or below) the horizontal axis represent transcripts that are significantly enriched (or depleted) in the HAX1 pulldown compared to their total abundance in cells.

#### 2.1.2. CRAC Results

CRAC (crosslinking and cDNA) analysis was performed in the HEK293FlpInTRex human embryonic kidney cell line modified to overexpress HAX1 after doxycycline induction. This cell line was used because it has been standardized for CRAC experiments. Two different cell lines were generated expressing a protein with a tag added at the 5′ or 3′ end of the HAX1 coding sequence and compared to the negative control cell line (experiments were performed in two replicates for each cell line). Pooled results for N- and C-tagged protein were considered for analysis (log2FC[CRAC_HAX1/CRAC_NC] > 0). Enrichment analysis (DAVID) revealed involvement in translation and ribosome assembly, including translation and cytoplasmic translation, rRNA processing, and ribosome biogenesis (Figure 3A, Appendix A). Enrichment values plotted against false discovery rate (FDR) for GO terms obtained by gene ontology enrichment analysis (Panther) revealed that similar terms as in RIP (involving translation, rRNA processing, and ribosome biogenesis) are of the highest significance (Figure 3B).

The transcript biotype analysis indicates a high proportion of rRNA and tRNA (respectively 54% and 17%), with mRNA representing 10% of CRAC targets (Figure 3C). Discrepancy with RIP results may result from methodical differences (UV crosslink, overexpression of the protein, see Discussion). Detailed analysis of the coverage of HAX1 reads along small (18S) and large (28S) ribosomal subunits shows high reads for the region of 1368–1407 in the large subunit, indicating a possible interaction site (Figure 3D and Appendix A).

The CRAC method enables the assessment of the enrichment in recurring sequence patterns (motifs) in an analyzed experimental dataset. The motif finding was performed using STREME (The MEME Suite; motif-based sequence analysis tools), resulting in a characterization of a guanine-rich motif for the C- and N-terminally tagged combined datasets (Figure 3E and Appendix A).

The analysis of the genomic position of the detected mRNA targets (269 best quality targets from the pooled C and N dataset) revealed the predominant binding of HAX1 to the coding sequence (CDS) region and the substantial number of targets located in both CDS and 5′ or 3′ UTRs. Only 11% of targets were mapped only in noncoding regions. Interestingly, introns plus CDS represent about 6% of targets, and half of these intronic targets encode snoRNA (Figure 3F, Appendix A).

#### 2.1.3. Comparison of the RIP-Seq and CRAC Results

In both approaches, the enriched GO terms encompassed translation, ribosome biogenesis, rRNA processing, and RNA processing in general. Enrichment in the same categories was detected for the group of 629 targets that overlapped on both screens, and these categories were also the most probable (Figure 4A–C), with a high proportion of transcripts encoding ribosomal proteins (including 33 proteins of large and small ribosomal subunits), ribosome assembly factors, and translation factors (Appendix A). These results point to the probable role of HAX1 in ribosome assembly and translation.

#### 2.1.4. Validation of a Selected Target by Microscale Thermophoresis

Microscale thermophoresis (MST) was performed for the in vitro transcript containing a fragment of the coding sequence of the RPL19 ribosomal protein. This target was selected due to its relatively high rank on the list of overlapping RIP and CRAC targets. The specific sequence was based on the CRAC identification of the binding region within exon3. The titration of the target RNA to a constant concentration of the fluorescently labeled HAX1 protein allowed for the determination of the ligand dissociation constants (Kd) and the systematic comparison of their binding affinity. The half maximal effective concentration (EC50) was calculated by the Hill fit model. The results reveal binding for the sense transcript, but not for the antisense negative control (Figure 4D and Appendix A).

### 2.2. HAX1 KO Affects the Expression Profile of HL-60 Cells

To assess the impact of HAX1 on the HL-60 transcriptome, two independent cell lines with HAX1 CRISPR/Cas9 knockout were generated (HAX1 KO#1, KO#2, Appendix A) and used in the RNA-seq experiment to compare expression profiles with HAX1 WT cells. Statistical significance was assigned only for genes differentially expressed in both *HAX1* KO cell lines vs. WT and in the same direction (*p*-value cutoff 0.05). Gene ontology analysis revealed that HAX1 knockout affects the expression profile of the HL-60 cells in several terms of the Biological Process (Figure 5A–C). Weighted analysis of the RNA-seq results (String 11, functional enrichment analysis for proteins ranked according to fold change [RNA-seq_*HAX1* KO/RNA-seq_WT]) allows to distinguish two main groups: GO terms linked to ribosome biogenesis, rRNA processing and translation (including mitochondrial translation), and terms linked to energy generation in mitochondria: respiratory electron transport chain and oxidative phosphorylation (details of GO analysis in Appendix A). *HAX1* knockout significantly affects the expression of 2344 genes (1158 upregulated and 1186 downregulated in KO). As indicated in Figure 5C, genes related to ribosome biogenesis, translation, and energy generation in mitochondria tend to be downregulated in *HAX1* KO cells.

### 2.3. Correlation Analysis of HAX1 Expression in Cancer Databases Reveals Differences in the Same GO Terms as Transcription Profiling in HL-60 Cells

The correlation of *HAX1* expression with other genes can be analyzed in expression databases created using high-throughput methods applied to patient samples from many different neoplasms. Neoplasms selected for the correlation analysis were chosen to correspond to the HL-60 cell line (leukemia) and another hematologic neoplasm (lymphoma) or common types of epithelial origin cancer (breast, cervical cancer).

Coexpression analysis was performed using cBioPortal for Cancer Genomics with TCGA data (The Cancer Genome Atlas, PanCancer Atlas) for the four neoplasms (Cervical Cancer—297 patients, Breast Cancer—1084 patients, Acute Myeloid Leukemia—200 patients, and Diffuse Large B-cell Lymphoma—48 patients). Gene lists with Spearman rank correlation coefficients were used in the String 11 gene ontology analysis (with Spearman coefficients as values/ranks) and the obtained Biological Process terms were plotted in Figure 6A–D, showing that the terms with the highest enrichment and the lowest false discovery rate are similar to the expression profiling obtained in HL-60 cells (RNA-seq). The most enriched and probable terms revealed by the weighted analysis of the genes whose expression significantly correlates with *HAX1* expression included those involved in translation (cotranslational protein targeting to membrane, mitochondrial translation, translational elongation, translational initiation, cytoplasmic translation), ribosome assembly, and rRNA processing, but also energy generation in mitochondria (respiratory electron transport chain, oxidative phosphorylation, cellular respiration), and RNA processing in general. These enrichments were observed in all neoplasms analyzed. For details, see the Appendix A.

Similar enrichments were also observed in other types of cancer (renal cell carcinoma, glioblastoma; data not shown).

### 2.4. Comparison of RIP-Seq and RNA-Seq Suggests That HAX1 Binding May Stabilize a Subset of Transcripts Involved in Ribosome Biogenesis

RIP-seq analysis and RNA-seq for *HAX1* KO/WT were performed in the same HL-60 cell line; thus, the comparison should reveal whether RNA binding has an effect on RNA stability.

The RNA-seq fold change (RNA-seq_*HAX1*_KO/RNA-seq_WT) distributions of the transcripts representing the HAX1 targets (RIP target subset) and the rest of the transcripts (non-RIP target subset) were compared (Figure 7A), showing that the changes in the distribution towards the downregulation of the RIP-associated transcripts in *HAX1* KO increase with the increasing probability of the HAX1-specific RNA interaction (lower FDR), suggesting that the RIP targets are less stable. The effect, although statistically significant, is not ubiquitous and does not affect all RIP targets equally, suggesting some other layers of regulation. Distribution analysis performed only for transcripts involved in ribosome biogenesis (red line) shows a much more substantial shift toward lower FC values (log2[RNA-seq_*HAX1*_KO/RNA-seq_WT]). Furthermore, the fraction of transcripts involved in ribosome biogenesis increases when the RIP results are analyzed with the same FDR cut-offs in the weighted analysis (String 11, Biological Process) with FC from RNA-seq as values (Figure 7B).

Interestingly, HAX1 binding seems to be limited to transcripts linked to translation/ribosome biogenesis and includes far fewer transcripts linked to the oxidative phosphorylation/respiratory electron chain, which were also significantly downregulated in *HAX1* KO, indicating a different mode of regulation for this other group of transcripts. Furthermore, mRNA RIP targets (RIP target subset) along with their respective FCs in RNA-seq were analyzed in weighted analysis (String 11, values, Biological Process), separately for the downregulated and upregulated transcripts, revealing an enrichment in ribosome biogenesis and rRNA processing (GO:0042254 and GO:0006364) for the group of downregulated transcripts, while the same analysis of the upregulated group did not reveal any enrichment (Figure 7C).

The possibility of the direct stabilization of transcripts by HAX1 binding was tested for two mRNAs: DHX37(encoding DEAH-box helicase 37, involved in ribosome biogenesis and translation initiation) and RRP7A (encoding ribosomal RNA processing 7 homolog A, predicted to be involved in rRNA processing and small ribosomal subunit assembly). The mRNA targets were selected from transcripts downregulated in *HAX1* KO and simultaneously from the results of the RIP/CRAC analysis as potential HAX1 targets with a binding region within the coding sequence of the transcript. Transcription was inhibited by Actinomycin D treatment and subsequent mRNA degradation was analyzed by qPCR in *HAX1* WT and *HAX1* KO cell lines, indicating faster degradation in the HAX1-deficient cell line (Figure 7D).

To test the possibility that the effect of HAX1 on the transcriptome may not be direct but may instead consist of influencing the transcription factors (TFs) and propagating the effect to the group of genes regulated by these factors, TFs potentially regulating the subset of genes for which the expression has changed in the *HAX1* KO were identified using the Enrichr package (Transcription, ENCODE and ChEA) and juxtaposed with the ranked list of RIP targets. TFs of the highest rank (FC, FDR) and the highest combined score provided by Enrichr are listed in Appendix A. YY1 and USF1 are the only TFs with FC > 2, but MYC, which has the highest combined score, although not highly ranked itself, should be considered because MYC-binding proteins are also among the RIP targets, with MYCBP of FC>3. Genes regulated by all these TFs are mostly downregulated in *HAX1* KO (Appendix A). However, the expression of all of these TFs is not changed at the mRNA level, as determined by RNA-seq, undermining the possibility of the indirect regulation via TFs.

### 2.5. HAX1 KO Affects the 40S:80S Ratio

To assess the physiological effect of HAX1 deficiency in the context of ribosome biogenesis, ribosome sedimentation in the sucrose density gradient was performed for *HAX1* WT and *HAX1* KO HL-60 cells. The ribosomal P/M (polysome-to-monosome) stoichiometry ratio did not reveal significant changes between the WT and KO cell lines, but the 40S to 80S ratio revealed that the peak corresponding to the small subunit increases in *HAX1* KO compared to WT (Figure 8A,B). The overall subunits:monosome ratio (40S + 60S:80S) is also significantly increased in *HAX1* KO compared to WT (Figure 8B).

## 3. Discussion

In this manuscript we report a comprehensive analysis of the HAX1–RNA interactome by two independent, high-throughput methods, which both suggest HAX1 involvement in the regulation of transcripts controlling translation, rRNA processing, and ribosome biogenesis. The two methods differ in complexity and specificity (Figure 1A), and the experiments were carried out in two different cell lines (HL-60 promyelocytic cell line, selected due to the strong effect of HAX1 inactivation in these cells, and HEK293 embryonic kidney cells with adrenal endocrine characteristics, selected for technical reasons). Nevertheless, both approaches yielded enrichment in similar GO terms in the Biological Process category (translation, rRNA processing, and ribosome biogenesis). The pool of overlapping targets obtained simultaneously by both methods also shows involvement in translation, rRNA processing, and ribosome biogenesis.

The distribution of RNA biotypes detected in both methods was different, with RIP targets predominantly consisting of mRNA and CRAC targets with a high proportion of rRNA and tRNA. These differences may stem from the fact that in CRAC: (1) the protein of interest was overexpressed, which should increase the abundance of nonmitochondrial protein able to interact with rRNA compared to endogenous RIP; (2) RNA targets were UV-crosslinked to it, which may shift the distribution towards less stable, transient interactions eliminated in RIP; (3) the background (negative control sample) is cleaner than in RIP-seq, and thus it may be easier to detect significant fold changes for very abundant transcripts (rRNAs, tRNAs, and snRNAs).

The next important question was if the expression of HAX1 RNA targets has changed after *HAX1* knockout. This comparison was made for HL-60 cells, since this cell line was used in both RIP and RNA-seq expression profiling for *HAX1* KO vs. WT. The results indicate a partial overlap between HAX1 RNA targets and mRNAs downregulated in *HAX1* KO and the overlap is related to transcripts involved in ribosome biogenesis and translation, and, only to a very small extent, to the transcripts involved in energy generation in mitochondria, which are also significantly downregulated in *HAX1* KO. This result suggests that only the subset involved in ribosome biogenesis and translation is regulated through direct HAX1 binding and the subsequent stabilization of the mRNA. Therefore, the other detected changes (especially the downregulation of a very important subset of transcripts involved in energy generation) must be therefore regulated by a different mechanism.

Further support for the hypothesis of the HAX1 transcript-stabilizing role is provided by the quantitative assessment of DHX37 and RRP7A mRNA degradation. The DHX37 RNA helicase is involved in ribosome biogenesis, including the formation of the central pseudoknot structure of the small ribosomal subunit [20]. RRP7A is also predicted to be involved in rRNA processing and assembly of the small ribosomal subunit. Both transcripts represent top RIP/CRAC targets, which are downregulated in *HAX1* KO cells. Quantification of the degradation of the DHX37 and RRP7A transcripts revealed more dynamic degradation in *HAX1* KO cells, suggesting stabilization by HAX1.

Analysis of the genomic position of the RNA targets obtained using the CRAC method revealed the prevalence of coding sequence (CDS) regions, which is not typical for the regulatory RNA sequence and is not consistent with the genomic position of previously characterized HAX1 binding regions (3′UTR). However, new high-throughput analyses demonstrated that binding to CDS is not as uncommon as previously thought and may play a role in the regulation of mRNA stability [21]. Interestingly, this stabilization should refer to the situation when mRNA is not actively translated, and thus it is not covered and protected by ribosomes and susceptible to endonuclease attack, as in the case of protein CRD-BP, which binds to c-myc mRNA, thus protecting it [22]. Furthermore, CDS binding was also observed for the FMRP protein and was related to the recruitment of the APP mRNA to processing bodies (P-bodies), which was proposed to restrict translation [23]. In line with this observation, we previously reported that HAX1 was observed to colocalize with the P-body marker, Dcp1 [12], pointing to its possible role in transcript stabilization during storage.

Interaction with one of the targets within CDS reported by both methods (RPL19) was confirmed by microscale thermophoresis, with the dissociation constants indicating relatively weak binding (K_d_ in a range of 0.1–0.2 μM, comparing, for example, to the strong FMRP interaction with N19 RNA with a K_d_ of 1 nM [24]). These values suggest a transient, regulatory interaction. Interestingly, the K_d_ values previously reported for HAX1 binding to 3′UTR are lower [8], indicating a different strength of interaction and, possibly, a different mode of binding for the CDS and 3′UTR regions. A similar phenomenon was also observed for the GLD-1 and FMRP proteins, involved in translation regulation [21].

Analysis of the possibility of indirect regulation mediated by transcription factors indicated that such regulation is improbable, since the expression of TFs itself is not changed in *HAX1* KO. However, this hypothesis cannot be totally dismissed, since TF-encoding transcripts may be differentially translated, or their protein product may be degraded in *HAX1* KO cells, resulting in differences in TFs at the protein level and subsequent changes in specific groups of transcripts regulated by those TFs.

Interestingly, the analysis of the correlation of expression with HAX1, performed for the TCGA database in four neoplasms (AML, DLBCL, breast cancer, cervical cancer), identified enrichment in the same biological processes as detected for *HAX1* KO vs. WT in the HL-60 cell line, indicating that these results are not cell line or neoplasm-specific, and thus further corroborating these findings.

The observed changes in expression related to ribosome assembly and translation are not huge, but reproducible, and refer to a relatively large group of transcripts and do not appear to be cell-type or neoplasm-specific. Relatively weak binding and small but reliable changes in expression suggest regulation via small, additive effects. It is an open question whether these effects can manifest more robustly in nonquiescent cells subjected to some kind of stress. This conjecture is supported by the reported changes in the location of HAX1 after stress, including nucleocytoplasmic shuttling [11], which could be linked to ribosome biogenesis and RNA binding in the nucleus/nucleolus. Furthermore, the abundance of rRNA observed as a potential target in the CRAC analysis suggests the possibility of a more direct involvement in ribosome biogenesis. The suggested binding site (Appendix A) maps within rRNA expansion segments, for which the function in ribosome biogenesis was proposed [25]. Thus, the involvement of HAX1 in ribosome assembly could encompass not only the regulation of the stability of mRNAs that encode ribosomal proteins and assembly factors, but also a direct interaction with ribosomal RNA. Interestingly, the possibility of the simultaneous regulation of translation by direct ribosome binding and controlling mRNA stability was described for the FMRP protein [26], already mentioned here for similar mode of binding and possible recruitment of transcripts to P-bodies.

To test the physiological consequences of *HAX1* KO on ribosome status, we performed ribosome sedimentation profiling, demonstrating a difference in the 40S:monosome ratio for HAX1-deficient cells. The shift towards free subunits in *HAX1* KO may indicate less efficient monosome assembly, resulting from lower expression of ribosomal assembly factors and ribosomal components.

In conclusion, we provide evidence for the involvement of HAX1 in ribosome assembly and translation, which represents a new finding. Previously, it was reported that HAX1 has an interaction with PELO, a protein involved in ribosomal rescue during ribosome stalling, but no mechanism for HAX1 involvement was proposed and no physiological effect was observed [27]. Recently, You et al. [28] demonstrated, among other things, that HAX1 levels correlate with ribosome formation and that HAX1 promotes the translation of the transcript encoding integrin subunit beta 6 in endothelial cells, partially matching the findings presented here.

Thus, the results presented suggest the possibility that HAX1 binds to the CDS of the nontranslated transcripts, protecting them from degradation, and that the main mRNA targets subjected to this regulation include transcripts involved in ribosome biogenesis.

Changes in ribosomal status and translation efficiency affect proliferation, which in turn can contribute (in opposite directions) to neutropenia and/or cancer. Indeed, the status of HAX1 has already been shown to affect proliferation [29], and we also observed this effect in our cell lines (data not shown). Further research should elucidate the exact role of HAX1 in maintaining translation efficiency, but the results presented here provide a starting point to explore these new and unanticipated possibilities.

## 4. Materials and Methods

### 4.1. Generation of Cell Lines

#### 4.1.1. HEK293FlpInTRex with Induced Overexpression of HAX1

Plasmid design and molecular subcloning: *HAX1* CDS was obtained by PCR and cloned into prepared vector (pcDNA5FRT TO) with a special gene coding tag (Protein A fragment, TEV protease cleavage site, and His-tag) in two orientations (tag at the 3′ or 5′ end of the *HAX1* coding sequence).

Cell line generation: HEK293-FlpIn cells (ThermoFisher Scientific) were transfected with an empty vector (negative control, NC) or plasmid with the *HAX1* cDNA with a tag at the 3′ end or 5′ end of the gene, and cotransfected with pOG44 vector, containing Flp recombinase, for Flp–FRT recombination. Transfection was carried out according to the manufacturers’ instructions (LipofectamineTM2000, ThermoFisher Scientific, Waltham, MA, USA). Cells were detached and seeded on 100 mm plates in a concentration enabling obtaining single colonies (selection: Blasticidin 15 μg/mL and Higromycin B 100 μg/mL) for 2 weeks. Single colonies were passaged to 24-well plates and tested by Western blot and qPCR with and without doxycycline induction (18–48 h).

#### 4.1.2. HL-60 HAX1 CRISPR Knockout

Plasmid design and molecular subcloning: two pairs (4 oligonucleotides) of small guide RNA (sgRNA) complementary to the *HAX1* gene were designed using the online bioinformatic tool (https://CRISPR.mit.edu, accessed on 15 August 2022). Each pair of sgRNA was introduced to the AIO-GFP plasmid that encodes the Cas9 nickase tagged with EGFP (D10A), as described in [30]. Briefly, each of the four oligonucleotides contained 5’ overhangs (forward: ACCG, reverse: AAAC) compatible with BbsI and BsaI restriction enzymes. The BbsI site present in AIO-GFP was utilized to introduce antisense (LC—left CRISPER) oligonucleotides and therefore generate AIO-GFP HAX1 LC1 and AIO-GFP HAX1 LC2 plasmids. To each respective plasmid, the second (sense, RC—right CRISPER) sgRNA of a given pair was introduced utilizing the BsaI site. As a result, two different constructs were generated, AIO-GFP HAX1 LCRC1 and AIO-GFP HAX1 LCRC2.

Cell line generation: HL-60 cell line was grown in RPMI1640 medium with L-glutamine (Biowest) and 10% Fetal Bovine Serum (Gibco) at 37 °C in a 5% CO_2_. Electroporation of 5 × 106 HL-60 cells was carried out with AIO-GFP HAX1 LCRC1 and AIO-GFP HAX1 LCRC2 using the CLB-Transfection™ Kit (Lonza, Austria) and the CLB-Transfection™ System (Lonza, Austria) with the default program 9 setting. Single transfected cells were sorted into separate wells of 96-well culture plate using BD FACSAria™ III (Becton Dickinson, Franklin Lakes, NJ, USA). Cells were cultured in RPMI for 14–21 days, colonies were propagated, and successful KO was validated by Western blot in four cell lines, two of which were used in experiments (Appendix A).

### 4.2. RIP-Seq

The HL-60 promyelocytic cell line (DSMZ, Braunschweig, Germany) was grown in RPMI1640 medium with L-glutamine (Biowest) and 10% Fetal Bovine Serum (Gibco) at 37 °C in a 5% CO_2_. Sample preparation: The experiment was carried out in five replicates. The EZ-Magna RIP RNA-Binding Protein Immunoprecipitation Kit (Millipore, 17–700 Sigma-Aldrich) was used according to the manufacturer’s protocol. A single freeze–thaw was employed to gently lyse the cells, as described by Keene et al. [31]. An amount of 30 × 106 cells per sample were collected by centrifugation at 966× *g*, 5 min at 4 °C, and washed two times with PBS containing protease inhibitors, resuspended in 200 μL of RIP lysis buffer (Millipore) containing protease and RNase inhibitors, incubated for 5 min on ice, snap-frozen in liquid nitrogen, and stored at −80 °C. Magnetic Beads Protein A/G (from Magna RIP Kit) were incubated overnight (4 °C) with 10 µg of anti-HAX1 rabbit polyclonal antibody (Thermo Fisher Scientific, Waltham, MA, USA; cat. PA5-27592) and Rabbit IgG (Millipore) as a negative control. The lysates from the previous step were quickly thawed, centrifuged at 14,000 rpm for 10 min at 4 °C, and 150 μL of supernatants were added to each antibody complex in the RIP Immunoprecipitation Buffer. The final volume of the immunoprecipitation reaction was 1.5 mL. A total of 10% of the sample was taken and stored as a total input. The lysate was incubated with antibody-coated beads for 4 h at 4 °C. After immunoprecipitation, the beads were washed 5 times with 1 mL of cold RIP Wash Buffer. The last, sixth wash was performed with 0.5 mL of wash buffer and 50 μL out of 500 μL of each beads’ suspension was taken to test the efficiency of immunoprecipitation by Western blotting. The remaining 450 μL of each suspension was collected with magnetic separator, immune complexes and input were eluted and treated with proteinase K (55 °C for 30 min). RNA was purified by extraction of phenol/chloroform followed by ethanol precipitation. The concentration of the precipitated RNA samples was checked using QuantiFluor RNA System (Promega), and the samples were used for library preparation and RIP-seq.

### 4.3. RNA-Seq

Sample preparation: HL-60 cell lines (WT and HAX1 KO#1 and #2) were grown to 3.5 × 106 cells in each culture (each cell line in four replicates). RNA was isolated using RNA PureLink Mini (Thermofisher, Waltham, MA, USA). Genomic DNA was removed from the samples using the TURBO DNA-free kit (Thermofisher, Waltham, MA, USA). RNA integrity was evaluated using Agilent RNA 6000 Nano Kit (Agilent Technologies, Waltham, MA, USA). RNA samples with a RIN score ≥ 9 were used for the preparation of cDNA libraries.

### 4.4. NGS Library Preparation and Sequencing (RIP-Seq, RNA-Seq)

The cDNA libraries were prepared using TruSeq™ Stranded Total RNA Library Prep Gold (Illumina, San Diego, CA, USA) according to the manufacturer’s procedure. The average size of the libraries was determined using the Agilent 2100 Bioanalyzer and High Sensitivity DNA Kit (Agilent Technologies, USA), while the concentration was assessed using the Qubit Fluorometer and dsDNA HS Assay Kit (Thermo Fisher Scientific, Waltham, MA, USA). Uniquely indexed libraries were pooled, mixed with Illumina PhiX Control v3 Library (1% of the total amount), and sequenced on HiSeq 1500 (Illumina) in Rapid Run Mode. Single-read sequencing (1 × 50 bp) and paired-end sequencing (2 × 100 bp) were performed for RIP-seq and RNA-seq, respectively.

### 4.5. RIP-Seq and RNA-Seq Data Analysis

#### 4.5.1. RIP-Seq

All experiments were performed in five replicates. Raw sequences were trimmed according to quality using Trimmomatic [32] (version 0.39) using default parameters, except MINLEN, which was set to 50. Trimmed sequences were mapped to the human reference genome provided by ENSEMBL (version grch38_snp_tran) using Hisat2 [33] with default parameters. Optical duplicates were removed using the MarkDuplicates tool from GATK [34] package (version 4.1.2.0) with default parameters, except OPTICAL_DUPLICATE_PIXEL_DISTANCE, which was set to 12000. The mapped reads were associated with transcripts from the grch38 database [35] (Ensembl, version 96) using HTSeq-count [36] (version 0.9.1) with default parameters, except for the stranded, which set to “reverse”. The variation was assessed by visual inspection of the first two components from the principal component analysis (PCA), which revealed correct clustering. FPKM was calculated with the fpkm function of the deseq2 package. Differentially expressed genes were selected using the DESeq2 package [37] (version 1.16.1). The fold change was corrected using apeglm. PCA, FPKM, and Deseq2 calculations were performed in the R environment (version 3.6).

#### 4.5.2. RNA-Seq

All experiments were carried out in four replicates. The DEGs selection was performed as described above.

### 4.6. CRAC (Crosslinking and Analyses of cDNAs)

#### 4.6.1. Sample Preparation

Stable established cell lines with HAX1 overexpression (coding protein tagged at the C or N terminal end, two replicates for each construct) were induced with 1 μg/mL doxycycline and incubated 18–48 h. After, the incubation cells were UV-crosslinked in Stratalinker 1800 (E = 400 mJ/cm^2^) at a wavelength 254 nm. Cells were lysed for 10 min on ice in lysis buffer (50 mM Tris-HCl pH 7.5, 300 mM NaCl, 1% NP-40, 5 mM EDTA, 10% glycerol, 5 mM β-mercaptoethanol). Lysates were spun at 4600 RPM, 4 °C for 5 min, and then filtered using syringe filter of 0.45 µm with PES membrane. IgG Sepharose was added to the lysates and incubated overnight at 4 °C with rotation. The beads were washed with IgG wash buffer (50 mM Tris-HCl pH 7.5, 800 mM NaCl, 0.5% NP-40, 5 mM MgCl_2_) and PNK wash buffer (25 mM Tris-HCl pH 7.5, 50 mM NaCl, 0.1% NP-40, 1 mM MgCl_2_). RNAs were trimmed on beads using 1 unit of RNAce-IT in 0.4 of PNK buffer for 7 min at 37 °C. To stop the reaction, the supernatant with RNaceIT was removed and the beads were resuspended in the room temperature denaturing elution buffer Ni-WBI (50 mM Tris-HCl pH 7.5, 300 mM NaCl, 1.5 mM MgCl_2_, 10 mM Imidazole pH 8.0, 0.1% NP-40, and 6 M guanidine hydrochloride). The elution was repeated one more time, both fractions were combined, and the Ni-NTA beads were added for overnight incubation at 4 °C. The NI-NTA beads were washed and transferred to Pierce columns. RNA was dephosphorylated with 8 U of Thermosensitive Alkaline Phosphatase (Promega) in supplied MultiCore buffer with 80 U RNAsin for 30 min at 37 °C. Beads were washed with Ni-WBI and PNK wash buffer. The 3′ linker ligation was performed overnight at 16 °C with 1 μM 3′ linker, 800 U of truncated T4 RNA ligase 2 K227Q (New England Biolabs) in supplied PNK buffer with RNAsin (Promega), and 10% (PEG8000). The beads were washed with WBI and PNK wash buffer. The RNA–protein complexes were radioactively labeled with 32P-γ-ATP (20 μCi) using 40 U T4 Polynucleotide Kinase (New England Biolabs) in the supplied PNK buffer for 30 min at 37 °C. The 5′ linker ligation was performed in the same reaction mixture by addition of the 5′ linker to the final 2.5 mM, nonradioactive ATP to final 1.25 mM, and 40 U T4 RNA ligase 1 for 8 h at 16 °C. After washing the beads with the Ni-WBI and PNK wash buffer, the RNA–protein complexes were eluted with elution buffer (50 mM Tris-Hcl pH 7.8, 300 mM NaCl, 150 mM Imidazole pH 8.0, 0.1% NP-40, 5 mM 2-mercaptoethanol) at room temperature for 5 min. Protein–RNA complexes were precipitated with 80% acetone in the presence of GlycoBlue at −20 °C overnight and spun for 20 min with max speed at 4 °C. Pellets were resuspended in LDS sample buffer (ThermoFisher, Waltham, MA, USA), DTT, and EDTA and denatured for 3 min at 90 °C.

#### 4.6.2. Autoradiography

Samples were resolved on 4–12% Bis-Tris NuPAGE gel at constant voltage (120 V) using NuPAGE MOPS SDS running buffer (Thermo Fisher Scientific, Waltham, MA, USA) and transferred to nitrocellulose membrane in NuPAGE transfer buffer using BioRad Protean wet transfer system at constant voltage (100 V) for 1 h. Exposition was performed at −80 °C overnight.

#### 4.6.3. RNA Isolation from Membrane

Bands corresponding to RNA crosslinked to the HAX1 protein (Figure 1D) were cut out and incubated with 450 μL Proteinase K buffer (50 mM Tris-HCl pH 7.8, 50 mM NaCl, 10 mM imidazole pH 8.0, 0.1% NP-40, 1% SDS, 5 mM EDTA, 5 mM 2-mercaptoethanol) and 200 μg Proteinase K for 2 h at 55 °C. The 3M sodium acetate pH 5.2 was added to final 10% and the RNA was extracted with 500 µL phenol:chloroform:isoamyl alcohol. After 5 min spin with a maximum speed at 4 °C, the aqueous phase was collected to a new tube and the RNA was precipitated with 3 volumes of ethanol in the presence of GlucoBlue.

#### 4.6.4. cDNA Library

Isolated RNA was incubated with SSIV reverse transcriptase (ThermoFisher) and a primer binding to 3′ linker. The cDNA was amplified using LA Takara Taq polymerase. PCR products were resolved on 3% Metaphor agarose gel (Lonza), and DNA fragments of sizes approximately 150–200 nt were isolated from the gel using Qiagen’s Gel Extraction Kit (Figure 1D). The cDNA library was sequenced on the Illumina MiSeq platform in Edinburgh Genomics (the University of Edinburgh).

### 4.7. CRAC Data Analysis

For CRAC experiments, all experiments (for N- and C-tagged protein) were performed in two replicates each. Fold change for each gene was calculated as a Log2 from the experiment/negative control ratio, normalized for total number of hits, and means from two replicates. NGS results were analyzed using algorithms: flexbar (preprocessing), tophat (genome mapping), and bedtools (analysis of genomic annotations). To identify HAX1 binding motifs, we extracted the coordinates of all HAX1 binding sites in mRNAs and created a control dataset by randomly placing the coordinates of these binding sites on the same mRNAs using shuffleBed. We then used the STREME sequence motif discovery algorithm (minimum motif length 4 nt, maximum length 8 nt) to identify enriched motifs [38] (https://meme-suite.org/meme/doc/streme.html, accessed on 15 August 2022).

Genomic position of HAX1 binding identification was done using UCSC Genome Browser on Human Feb. 2009 (GRCh37/hg19) Assembly.

### 4.8. Gene Ontology and Correlation Analysis

Enrichment plots for RNA-seq data were generated by GSEA (gene set enrichment analysis, [39,40]) software. Table of transcripts, identified during the RNA-seq experiment, with an associated number of counts per transcript (data obtained with HTSeq-count), was used as input data. Number of permutations was set to 1000 and permutation type was set to “gene_set”. Gene set database (version 7.2) used for analysis is included in the respective enrichment plots titles. The other gene ontology enrichment analyses were performed using packages: Gene Ontology resource (http://geneontology.org, accessed on 15 August 2022); ([41] Gene Ontology 2021), DAVID Functional Annotation Tool, DAVID Bioinformatics Resources, NIAID/NIH (https://david.ncifcrf.gov/home.jsp, accessed on 15 August 2022) [42,43], and String 11 Functional Enrichment Analysis for proteins with values/ranks (https://string-db.org, accessed on 15 August 2022) [44]. Transcription factors were analyzed using Enrichr ENCODE and ChEA Consensus TFs from ChIP-X (https://maayanlab.cloud/Enrichr/, accessed on 15 August 2022) [45,46].

Correlation analysis with high-throughput data accumulated in TCGA (The Cancer Genome Atlas) database was performed using cBioPortal for Cancer Genomics, a platform for exploring multidimensional cancer genomic data [47,48] (https://www.cbioportal.org/, accessed on 15 August 2022 ).

### 4.9. Transcription In Vitro

Transcription in vitro was performed for 140 nt RPL19 fragment of CDS from exon 3 (primers: FW 5′-GGTGCATTATGCTTTCCCAGGTCAG-3′, REV 5′-CTATGCCCATGTGCCTGCCCTTC-3′) cloned into pGEM-T Easy vector in sense and antisense orientation. M13 fwd and rev primers were used in the PCR reaction for template generation. MEGAscript T7 transcription Kit (ThermoFisher Scientific) was used for in vitro transcription with T7 RNA polymerase, according to the manufacturers’ protocol. Transcripts were purified using MEGAclear Kit (ThermoFisher Scientific, Waltham, MA, USA).

### 4.10. Microscale Thermophoresis (MST)

MST experiments were performed using Monolith NT.115 (NanoTemper Technologies GmbH, Munich, Germany). Purified HAX1 protein (Proteintech, Ag27244, fused with His-tag) was labeled with RED-NHS 2nd Generation dye according to the supplied labeling protocol Monolith NT™ Protein Labeling Kit. A series of dilutions of ligand RNA (sense and antisense transcript) were prepared using buffer solution containing PBS with 0.2% Tween-20. The solution of labeled protein was mixed 1:1 with different concentrations of RNA strand, yielding a final concentration of 50 nM of the protein and the ligand in a range of final concentrations between 10.8 µM and 0.000328 µM. After 5 min of incubation, the NT.115 premium capillaries (NanoTemper Technologies, Munich, Germany) were filled with the RNA/protein solution and thermophoresis was measured at an LED power of 100% and an MST power of 60% at RT. Each operation was controlled using the MO control software. The Kd was determined by nonlinear fitting of the thermophoresis responses and EC50 was determined by Hill fitting model using the MO Affinity Analysis v2.3 for both types of the calculations.

### 4.11. Western Blot

Protein extracts were heat-denaturated (95 °C) in Laemmli buffer (50 mM Tris/HCl, 0.01% Bromophenol Blue, 1.75% 2-mercaptoethanol, 11% glycerol, 2% SDS) and separated by 10–12% SDS/PAGE electrophoresis. Proteins were transferred to Immobilon-P PVDF membrane (Merck Millipore, Burlington, MA, USA). The membranes were incubated for 1h using 5% low-fat milk solution in 1X TBS (50 mM Tris-Cl, pH 7.5, 150 mM NaCl) as a blocking buffer, and then overnight at 4 °C in the same blocking solution containing one of the following antibodies: anti-HAX1 (rabbit, Proteintech 11266-1-AP) or anti-RPL26 (rabbit, 1:5000, Abcam, ab59567). After washing (3 × 10 min in TBS), membranes were incubated for 2 h at room temperature with the adequate HRP-conjugated secondary antibody: goat anti-rabbit IgG (1:5000, Abcam, GB; cat. 97051) or goat anti-mouse IgG (1:10,000, Abcam, GB; cat. ab97023). Membranes were developed using HRP detection kit WesternBright Quantum (Advansta, San Jose, CA, USA; cat. K-12042).

### 4.12. qPCR

Quantitative PCR was performed as described [11]. Briefly, stable cell lines with HAX1 WT and HAX1 KO were subjected to Actinomycin D treatment (10 μg/mL). Cells were harvested in a designated time points and used for total RNA preparation (PureLink RNA mini kit; Invitrogen), followed by the treatment with recombinant DNase I (Roche). An amount of 1 μg of the obtained RNA was used for cDNA synthesis using Superscript III (Invitrogen). The cDNA was quantified by quantitative PCR on an ABI Prism 7500 real-time PCR system using Power SYBR Green PCR Master Mix (Applied Biosystems, Life Technologies, Carlsbad, CA, USA) and primers amplifying a fragment of DHX37 transcript (forward 5′- CCCGATATCGAGAAAGCCTGG-3′; reverse 5′-CGTCCAGCACGTGAGATGAA-3′), RRP7 transcript: (forward 5′- TTCTCGTCACAAGGCACAGG-3′; reverse 5′-GAAGGGCCACACCTAAGTCC-3′) and, as a reference, ACTB transcript (forward 5′-AGCCTCGCCTTTGCCGA-3′; reverse 5′-GCGCGGCGATATCATCATC-3′). The ΔΔ CT method was used for calculating mRNA expression levels.

### 4.13. Sucrose Gradient Centrifugation

HL-60 cell lines (WT and HAX1 KO#1 and #2) were grown and subcultured until achieving 6 T-75 or 3 T-175 flasks with cells at a density of 1 × 106/mL. Cells were treated with cycloheximide (100 μg/mL) at 37 °C for 10 min, harvested by centrifugation for 5 min at 500× *g*, 4 °C, and washed 3 times with ice-cold PBS supplemented with 100 μg/mL cycloheximide. After final wash and complete removal of PBS, the cell pellet was resuspended in 0.75 mL of lysis buffer A (10 mM Tris-HCl pH 7.4, 12.5 mM MgCl2, 100 mM KCl, 0.5% Triton X-100 reduced, 2 mM DTT, 100 µg/mL CHX, 200 units SUPERaseIn™ RNase Inhibitor (20 U/μL; ThermoScientific), and cOmplete EDTA-free protease inhibitor (Roche)). Cells were lysed by thorough pipetting and incubation for 15 min at 4 °C on a rotating wheel. Lysates were aspirated into 1 mL syringe, passed through a 26G needle seven times, and then centrifuged for 10 min at 16,000× *g*, 4 °C. RNA concentration in clarified cytoplasmic extracts thus obtained was measured using Nanodrop 2000c (ThermoScientific). A total of 14–20 OD260 units of cytoplasmic extract in 500 μL of lysis buffer was layered on top of 10–50% linear sucrose gradients, prepared using ÅKTA Purifier FPLC system and 0.22 µm-filtered sucrose solutions in polysome buffer (20 mM Hepes-KOH pH 7.4, 12.5 mM MgCl2, 100 mM KCl; 2 mM DTT, 100 µg/mL CHX, and cOmplete EDTA-free protease inhibitor), and ultracentrifuged for 3 h 15′ at 36,000 rpm, 4 °C in SW-41Ti rotor (Beckman Coulter). Subsequently, 0.5 mL fractions were collected from gradient by pumping 60% sucrose solution in polysome buffer to the bottom of tubes, and OD254 was monitored on an ÅKTA Purifier.

## Figures and Tables

**Figure 1 cells-11-02943-f001:**
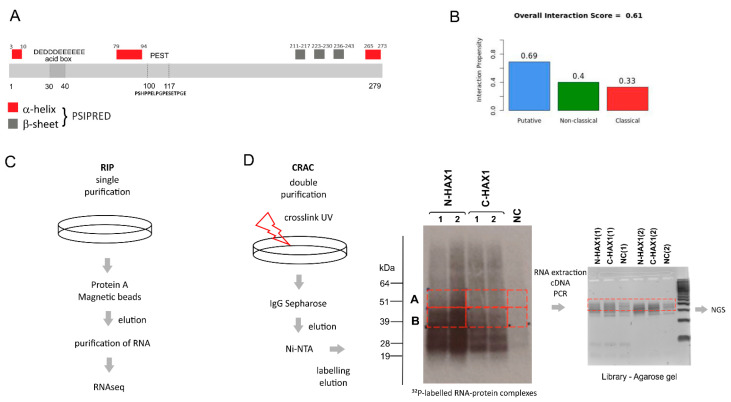
Experimental design of HAX1–RNA interactome studies. (**A**) HAX1 amino acid sequence with characteristic features and regions (acid box, PEST domain, structural elements predicted using PSIPRED (PSI-blast-based secondary structure PREDiction)). (**B**) CatRapid prediction of HAX1 RNA binding propensity classifies it as a putative RBP, but binding does not occur via classical elements. (**C**) Schematic of RIP experiments: no crosslink, single purification step with HAX1-specific antibody. (**D**) Schematic of CRAC experiments, sequential steps: UV crosslink of RNA targets, double purification with Sepharose and NiNTA resin, isotope labeling and extraction of the RNA–protein complexes from polyacrylamide gel, RNA extraction, reverse transcription, PCR, and next generation sequencing of the RNA targets.

**Figure 2 cells-11-02943-f002:**
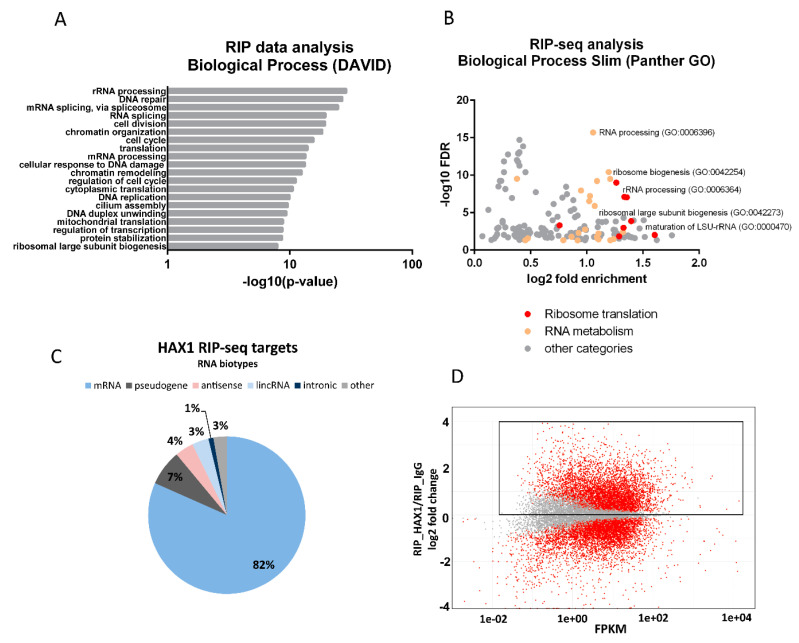
Identification of HAX1-associated RNA targets by RIP-seq analysis in HL-60 cells. (**A**) Functional annotation enrichment of HAX1 target genes. Enriched gene ontology terms (DAVID: Biological Process) presented in bar plot, the length of each bar corresponds to statistical significance of the enrichment (−log10 *p*-value). The top twenty terms are presented. (**B**) Enriched Biological Process terms (Panther GO) plotted with fold enrichment (log2) against False Discovery Rate (−log10). The most reliable results are in the upper right corner. Categories associated with ribosome biogenesis, rRNA processing, and translation are marked red. Categories associated with RNA metabolism (other than rRNA) are marked light orange. Detailed description of GO terms in Appendix A. (**C**) Distribution of RNA classes among HAX1 RIP targets. (**D**) MA plot showing RIP targets counts (X-axis) calculated as fragments per kilobase per million (FPKM); Y-axis: log2FC(RIP_HAX1/RIP_IgG), red—significant, grey—not significant. The internal frame indicates positive log2 FC (physiologically relevant as possible binding targets).

**Figure 3 cells-11-02943-f003:**
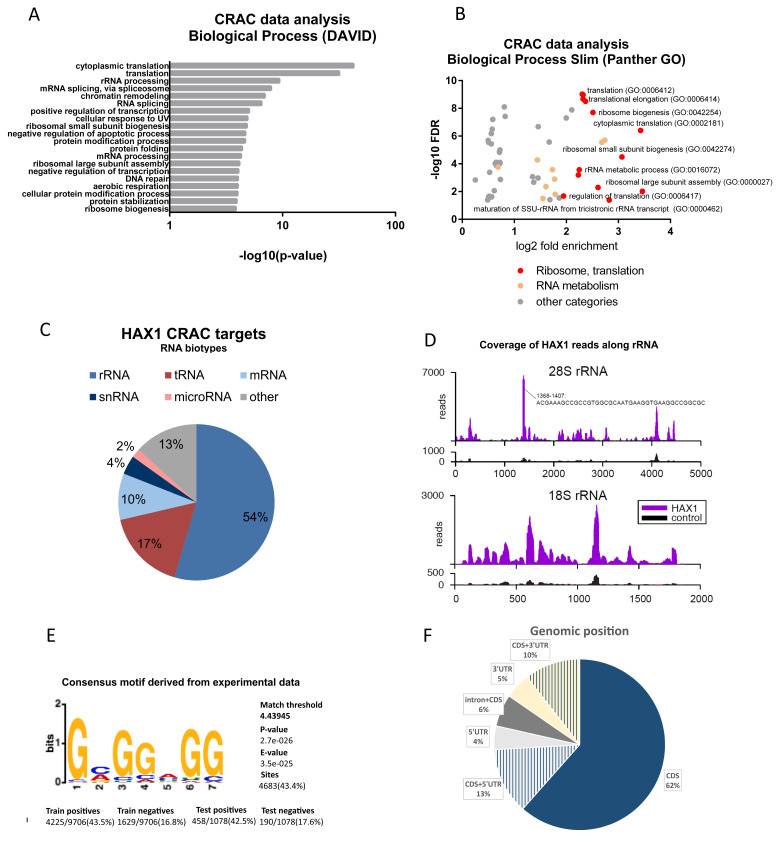
Identification of HAX1-associated RNA targets by CRAC analysis (HEK293 cells). (**A**) Functional annotation enrichment of HAX1 target genes (results from the pooled N- and C-tagged protein approach). Enriched gene ontology terms (DAVID: Biological Process) presented in bar plot, the length of each bar corresponds to statistical significance of the enrichment (−log10 *p*-value). The top 20 terms are presented for each category. (**B**) Enriched Biological Process terms (Panther GO) plotted with fold enrichment (log2) against false discovery rate (−log10). The most reliable results are in the upper right corner. Terms associated with ribosome biogenesis, rRNA processing, and translation are marked red. Terms associated with RNA metabolism (other than rRNA) are marked in light orange. Detailed description of GO terms in Appendix A. (**C**) Distribution of RNA species in HAX1 targets. (**D**) Coverage of HAX1 CRAC reads along small (18S) and large (28S) ribosomal subunits. Reads for negative control shown below. (**E**) Consensus motif identified by STREME in the HAX1-targets dataset (combined experimental data from C- and N-tagged HAX1). The E-value is the *p*-value multiplied by the number of motifs reported by STREME. (**F**) Genomic position of HAX1 binding in CRAC-identified targets (269 targets overlapping in C- and N-tagged datasets) established using UCSC Genome Browser on Human Feb. 2009 (GRCh37/hg19) Assembly.

**Figure 4 cells-11-02943-f004:**
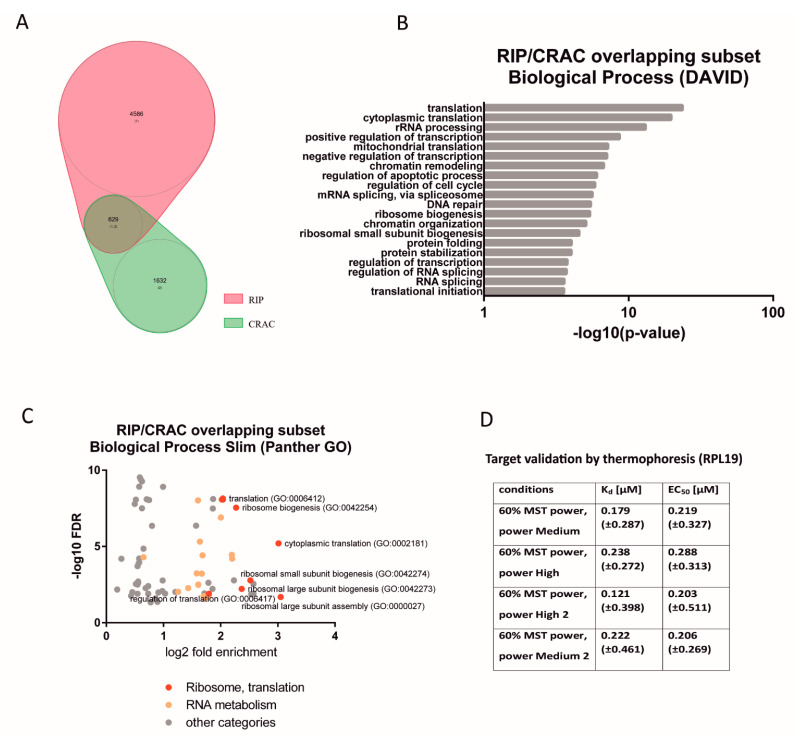
Characterization of the overlapping CRAC–RIP subset of HAX1 mRNA targets. (**A**) Overlapping targets in RIP and CRAC_C (629 mRNAs) represent, respectively, 13.7% and 38.5% of the overall results. Hypergeometric test for these overlap gives *p*-values of 9.8e–42 (total gene set: 15220). (**B**) Functional annotation enrichment of the RIP/CRAC overlap of HAX1 target genes (DAVID, top 20 terms) maintains a high enrichment in terms associated with translation, RNA and rRNA processing, and ribosome biogenesis (Biological Process) (**C**) Enriched Biological Process terms found for the overlap subset (Panther GO) plotted with fold enrichment (log2) against false discovery rate (−log10) reveal that terms associated with translation and ribosome biogenesis are the most probable. (**D**) Target validation by microscale thermophoresis (MST) performed for the RPL19 in vitro transcript (sense). From 33 transcripts encoding ribosomal proteins present in the overlapping RIP/CRAC target list, a fragment of the RPL19 coding sequence was selected for validation. The interaction with purified and the fluorescently labeled HAX1 protein was confirmed in 4 independent measurements by thermophoresis under different conditions (MST power medium to high). For each measurement, the dissociation constant (Kd) and half maximal effective concentration (EC50) were assigned. No interaction was observed with antisense transcript (negative control).

**Figure 5 cells-11-02943-f005:**
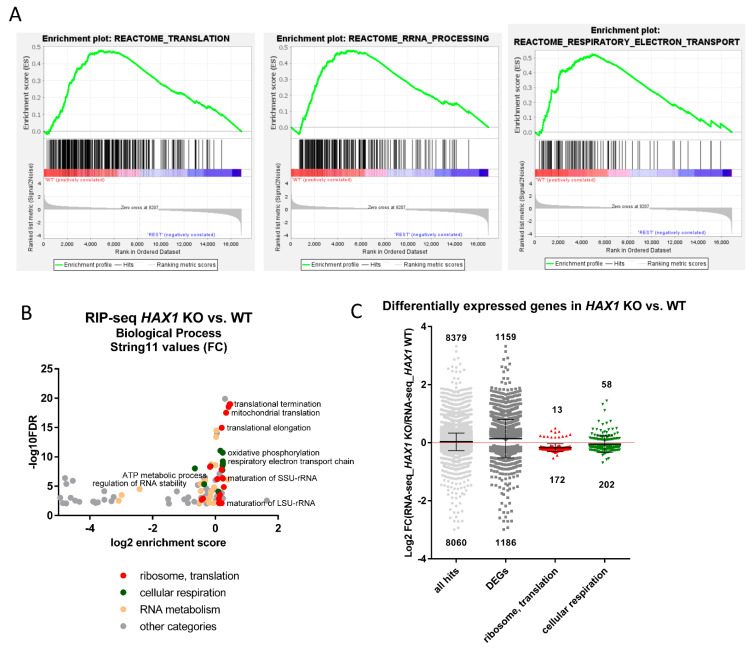
*HAX1* KO affects the expression profile in HL-60 cells. (**A**) Enrichment plots generated by GSEA analysis for the chosen categories (translation, rRNA processing, respiratory electron transport) (**B**) Enrichment in GO Biological Process terms for genes differentially expressed in *HAX1* KO vs. *HAX1* WT (String 11 weighted analysis with log2[RNA-Seq_*HAX1*_KO/RNA-Seq WT] as values) indicates participation in translation (including mitochondrial translation), rRNA processing/ribosome biogenesis, and energy generation in mitochondria. Detailed description of GO terms in Appendix A. (**C**) Scatter plot showing differentially expressed genes in *HAX1* KO. Of the total 1186 transcripts significantly downregulated in KO, 172 are associated with translation and 202 with energy generation, while from the 1158 transcripts significantly upregulated in KO, 13 and 58, respectively, are associated with these categories. The *p*-value cutoff: 0.05. A total of 14 outliers were omitted. The statistical significance of the downregulation of the transcripts involved in ribosome biogenesis and translation and cellular respiration was assessed by the Chi-Square test and was shown to be high (*p*-value = 5.5 × 10^−20^ and 1.2 × 10^−5^, respectively). For the other two groups (all mapped results and all DEGs), there was no statistical significance associated with downregulation.

**Figure 6 cells-11-02943-f006:**
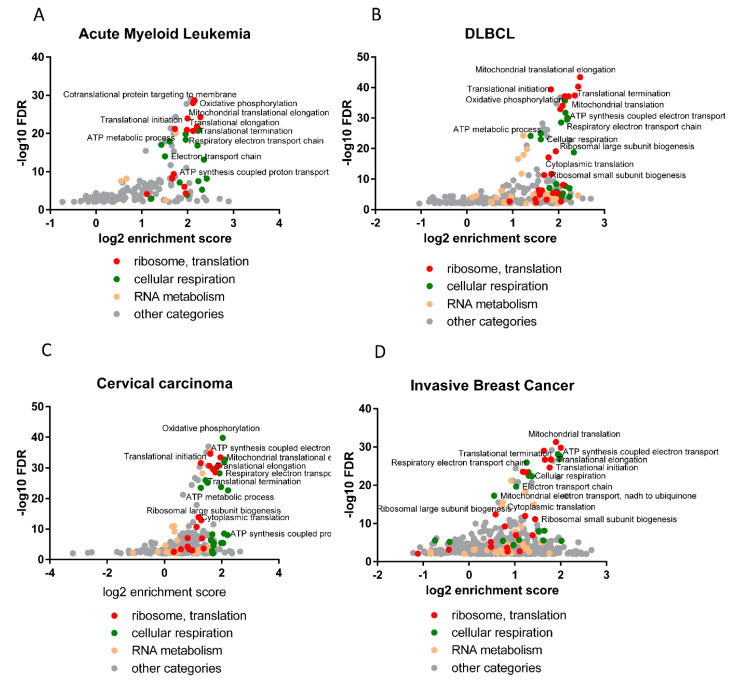
The subset of genes whose expression correlates with *HAX1* in the four different neoplasms exhibits enrichment in the same biological processes as shown for *HAX1* KO. Correlation analysis was performed using cBioPortal Cancer Genomics. Clinical data obtained from the TCGA database. Enrichment analysis was performed using String 11 with correlation coefficients as values. (**A**) Acute myeloid leukemia (AML, 200 patients); (**B**) Diffuse large B-cell lymphoma (DLBCL, 48 patients); (**C**) Cervical cancer (297 patients); (**D**) Invasive breast cancer (1084 patients) C. Detailed description of genes, Spearman coefficients and GO terms in the Appendix A.

**Figure 7 cells-11-02943-f007:**
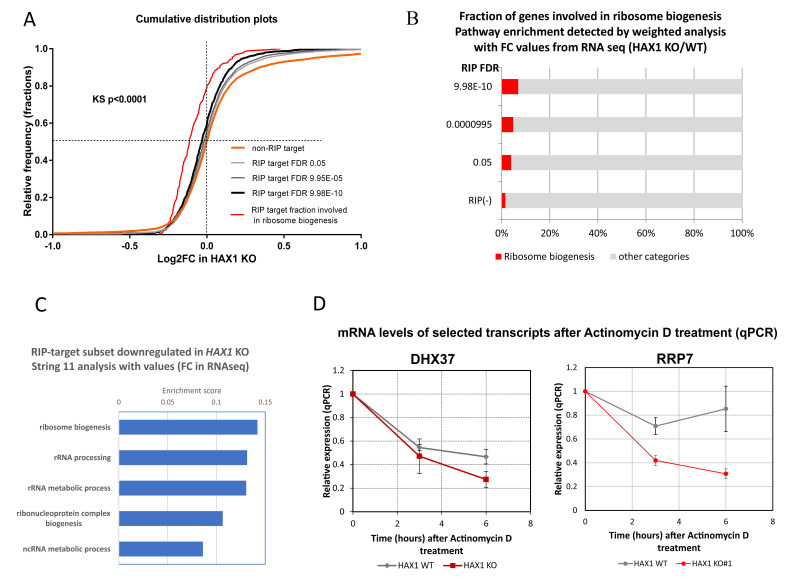
HAX1 binding to RNA targets may be responsible for changes in the expression profile (**A**). Comparison of the distribution of FC changes caused by HAX1 knockout (data obtained in RNA-seq) in the subset of transcripts present between targets identified by RIP (RIP target) and not identified by RIP (non-RIP target). Three different RIP target subsets were compared with cut-offs determined according to the decreasing FDR, showing a shift towards lower FC values (RIP_HAX1/RIP_IgG) in the subsets with lower FDR. Red line: a subset of transcripts involved in ribosome biogenesis, identified by String 11 weighted analysis for RIP-target transcripts (RIP_HAX1/RIP_IgG FC as values). Significance calculated by the Kolmogorov–Smirnov test (**B**). The fraction of transcripts involved in ribosome biogenesis increases with decreasing FDR for the subset of RIP targets. Pathway enrichment calculated in String 11 weighted analysis with FC values from RNA-seq (RNA-seq_*HAX1*_KO/RNA-seq_WT). (**C**) Enrichment in GO terms assessed by String 11 weighted analysis with FC values from RNA-seq for a subset of downregulated transcripts. Analogous analysis for the upregulated subset produced no results. (**D**) The degradation of DHX37 and RRP7 mRNAs is more dynamic in *HAX1* KO cells. Cells were treated with Actinomycin D (10 μg/mL). Relative expression was quantified by qPCR at designated time points. The experiment was carried out in several biological repeats and evaluated by the t-test (DHX37, n = 4, *p*-value: 0.026, 6 h, RRP7A n = 3, *p*-value: 0.035, 3 h). Error bars: SEM.

**Figure 8 cells-11-02943-f008:**
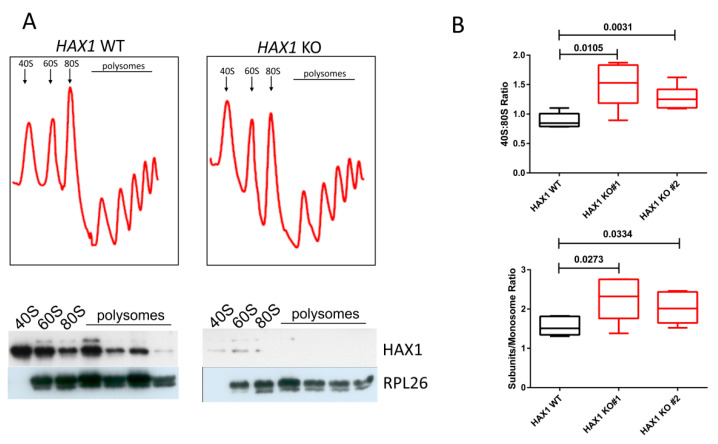
*HAX1* KO affects the subunit:monosome ratio. (**A**) Representative absorbance profiles at 254 nm of sucrose gradient (10–50%) sedimentation of HL-60 cell extracts. The three leftmost peaks include ribosomal subunits (40S and 60S) and nontranslating monosomes (80S). The remaining peaks represent polysomes. Western blots for HAX1 and RPL26 ribosomal protein (large subunit) performed for the corresponding fractions are presented under the profiles. (**B**) Quantitative analysis of sedimentation profiles indicates that free subunits are more abundant in *HAX1* KO in relation to the monosome (upper graph: 40S:80S ratio, lower graph: subunits/monosome ratio). The area under each peak was quantified using ImageJ (n = 6 in each sample), statistical significance determined with an unpaired *t*-test. Analysis by one-way ANOVA and Tukey multiple comparison test also show statistically significant difference for the 40S:80S ratio (*p*-value: 0.003 and 0.0486 for *HAX1* KO #1 and #2, respectively) and for the subunits:monosome ratio (*p*-value: 0.0296 for *HAX1* KO #1, for *HAX1* KO #2, *p*-value: 0.15).

## Data Availability

HAX1 targets genomic position available at UCSC Genome Browser (http://genome-euro.ucsc.edu/s/Grzegorz%20Kudla/Ewelina_HAX_MCPIP, accessed on 15 August 2022). Source data for the coverage of CRAC reads: gkudla@bifx-rta:/homes2/gkudla/Solexa/GK/Ewelina_Macech/20181219_Ola_sequencing/mapping/motifs_EMC1_EMC2_combined. NGS data have been deposited with the Gene Expression Omnibus (GEO) functional genomics data repository, accession numbers: GSE189611, GSE189609.

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
