# Peer review of "The RNA-Binding Landscape of HAX1 Protein Indicates Its Involvement in Translation and Ribosome Assembly"

_cells, 2022, doi:10.3390/cells11192943_

Round 1
Reviewer 1 Report
Using next-generation sequencing approaches, Balcerak et al. characterize the protein HAX1 at the transcriptome level. This protein is involved in disease, but its lack of homology to known proteins and the absence of known structured domains make it challenging to understand how it functions. The findings by Balcerak et al. suggest a role for HAX1 in ribosome biogenesis and translation.
The research design is appropriate to answer the question. HAX1 is an uncharacterized RNA-binding protein. The authors help close this gap of knowledge by carrying out RIP-seq and CRAC to characterize the RNA-binding targets of HAX1. They also carry out biophysical experiments to characterize binding, as well as sucrose gradienst, which reveal an effect of HAX1 on the partition of ribosomes and their subunits.
Most of our comments below pertain to improving the flow and readability of this manuscript, so that readers can mre readily grasp the relevance of the results:
- lines 48-50: further details about the mentioned 3'UTRs and hairpin structures could help readers.
- lines 56-59: further details explaining what PEST sequence and BCL2-like domains are would be helpful to the reader. Also, how long is HAX1?
- lines 75: what kind of information do these RNA binding prediction programs rely on to predict RNA binding in absence of secondary and tertiary motifs homolog to existing structures? What do the scores correspond to?
- line 87: A more detailed explanation of the displayed gels in the Figure 1 legend would be useful.
- lines 88-89: no experiment is "straightforward", particularly to people outside the immediate field, so such qualitative statements do not add much to writing instead "RIP-seq comprises one purification step..."
- lines 105-106: “So the analysis of the role of HAX1 in these cells seems to be the most physiological relevant” reads like a relatively weak statement, could the authors rephrase?
- line 150: it looks like Figure S2 should be cited in the RIP-seq section and not the CRAC section?
- lines 154-156: how do the authors explain that the majority of targets according to RIP-seq are mRNAs, but according to CRAC they are tRNAs and rRNAs? Which one to believe or why are there such differences? The authors have a subsequent section (lines 187-194) on a comparison of the datatsets with a nicely illustrated FIg 4 but they do not address these points specifically. The readers has some of an interpretation later in the discussion (lines 388-392), but would benefit from a pointer earlier in the text.
- line 158: show coverage for a non-rRNA sequence in panel 3D (or in the supplementary materials) as negative control
- lines 164-169: do these CRAC hits to mRNAs match mRNAs identified using RIP-seq?
- lines 212-220: the raw MST traces and fits need to be shown to the reader at least in the supplemental. This includes the negative control antisense RNA. REference to FIgure 4E in the text should be corrected to Figure 4D.
- line 365: legend should include a description of the gels shown under the profiles.
- lines 427-430: the interaction is described as weak, but related to what? and why are these values typical of IDR, could the authors add points of comparison with references? (as they do in the last section of this same paragraph)
Comments to methods:
- lines 489-499: which vectors were used for cloning HAX1?
- lines 495-497: What does it mean that cells were seeded into 100 mm plates to grow single colonies? Could the authors add some more details about what they did here? The symbol for "microg" is missing in the units. Selection was probably carried out with 15 ug/ml Blasticidin and 100 ug/ml Hygromycin. Also, was this two different selections or were both of these antibiotics used at the same time?
- lines 501-520: What size plate was used for transfection and at what confluency? How was the transfection carried out?
General comments:
- please define abbreviations the first time they're used: HAX1, PEST, BCL2, CRAC, DAVID, etc. Provide further context/background as well to better understand the results, as detailed above.
- italicize organism names, like Homo sapiens.
- super/subscripts missing throughout the experimental section (106, MgCl2, cm2, etc)
Author Response
Please, see attachment. Authors answers/comments in red.

Reviewer 2 Report
The manuscript “The RNA-binding landscape of HAX1 protein indicates its involvement in translation and ribosome assembly” is a carefully conducted and well-written study about the role and the possible mode of action of the still insufficiently understood and unusual RBP HAX1. The authors use two complementary interactomic techniques (RIP-seq and CRAC) to comprehensively characterise the ensemble of HAX1 in vivo RNA ligands. They parallel these findings by analysing the transcriptome changes upon HAX1 KO. Both analyses zero in on the especially statistically strong group of targets related to ribosome biogenesis and translation, including most ribosomal proteins. Comparing the HAX1 interactome with the transcriptomic data, the authors suggest that part of the observed gene expression changes could be directly caused by HAX1 stabilising binding to its target RNAs. Although these gene expression shifts are small, the authors support their claim by sound statistical analyses, pathway enrichment data, binding motif identification, in vitro binding assays, and stability measurements. Finally, they make an interesting observation that the ribosomal profile features an unusual increase in 40S peak, reinforcing the impression that HAX1 is somehow connected to ribosome biogenesis. I have no reserves about the publication of this interesting study. Below I give just a couple of minor points which may further improve or clarify the presentation of the data.
1. It seems that the interactomic data presented in this paper do not fully agree with earlier observations regarding HAX1 ligands (vimentin, DNA-Pol β, integrin β6 mRNAs). Could the authors comment on this?
2. Figure 1D: The legend does not permit to understand what exactly is shown on the blot and gel images. Could the authors provide more detail/guidance through this panel for audience less versed in CLIP-like techniques?
3. For Figure 4A, a hypergeometric test could be calculated to assess the significance of the overlap.
4. Figure 7D: Define the centrality measure and the bars.
5. Figure 8A: The legend says nothing about the blots.
6. Ll. 388-392: I suspect an additional reason for the difference between the RIP-seq and CRAC results. The background (i.e. the negative control sample) of CLIP-family methods is usually much cleaner than in RIP-seq (where it usually closely reproduces the total cellular transcriptome). This is why it may be easier to detect significant fold changes for very abundant transcripts (like rRNAs, tRNAs and snRNAs) in CLIP/CRAC than in RIP-seq.
7. Ll. 448-449: “Small effects” is one of possible hypotheses. However, modest changes in stability often indicate a different molecular mechanism which may have more to do with regulation of translation or RNA localisation.
8. The authors could reorder the Supplementary figures as they are referred to in the text (currently, S2, S3, S4, S1, S5).
9. L. 73: “analyses” (noun).
10. L. 220: “Figure 4D”.
11. Ll. 225-226: Somewhat unclear sentence. The data file S4 actually contains all fold changes and P-values for either replicate alone and for both the replicates together.
12. L. 323 and 339: By “more dynamic” do the authors mean simply “faster”?
13. Ll. 340, 710 and 712: A symbol seems to be lost in “10 g/ml” and “1 g”.
14. L. 375: For “not significant” provide the exact P-value.
15. L. 447: The term “persistent” is unsuitable in this context as it suggests constancy in time. Maybe “pervasive” instead?
16. L. 480: “translation efficiency affect proliferation”.
17. L. 490: “with a special tag”.
18. L. 708: The reference number is missing.
19. Throughout Materials and Methods: The authors should pay more attention to subscripts and superscripts. In some instances, this deforms too much the intended expression (e.g. in ll. 527, 700, 724, 732, 733).
Author Response
Please see attachment. Athors answers/comments in red.
